# Mutations in chikungunya virus nsP4 decrease viral fitness and sensitivity to the broad-spectrum antiviral 4′-Fluorouridine

Peiqi Yin[1], Elizabeth B. Sobolik[2¤], Nicholas A. May[3], Sainan Wang[4], Atef Fayed[1], Dariia Vyshenska[2], Adam M. Drobish[3], M. Guston Parks[3], Laura Sandra Lello[4], Andres Merits[4], Thomas E. Morrison[3], Alexander L. Greninger[2], Margaret Kielian[1]*

**1** Department of Cell Biology, Albert Einstein College of Medicine, Bronx, New York, United States of America, **2** Virology Division, Department of Laboratory Medicine and Pathology, University of Washington Medical Center, Seattle, Washington, United States of America, **3** Department of Immunology and Microbiology, University of Colorado School of Medicine, Aurora, Colorado, United States of America, **4** Institute of Bioengineering, University of Tartu, Tartu, Estonia

¤ Current address: David Rockefeller Graduate Program in Bioscience, Rockefeller University, New York, New York, United States of America

* margaret.kielian@einsteinmed.edu

**Data Availability Statement:** All data supporting the findings of this study are available within the paper and its supplemental data and source files.

## Abstract

Chikungunya virus (CHIKV) is an arthritogenic alphavirus that has re-emerged to cause large outbreaks of human infections worldwide. There are currently no approved antivirals for treatment of CHIKV infection. Recently, we reported that the ribonucleoside analog 4′-fluorouridine (4′-FIU) is a highly potent inhibitor of CHIKV replication, and targets the viral nsP4 RNA dependent RNA polymerase. In mouse models, oral therapy with 4′-FIU diminished viral tissue burdens and virus-induced disease signs. To provide critical evidence for the potential of 4′-FIU as a CHIKV antiviral, here we selected for CHIKV variants with decreased 4′-FIU sensitivity, identifying two pairs of mutations in nsP2 and nsP4. The nsP4 mutations Q192L and C483Y were predominantly responsible for reduced sensitivity. These variants were still inhibited by higher concentrations of 4′-FIU, and the mutations did not change nsP4 fidelity or provide a virus fitness advantage in vitro or in vivo. Pathogenesis studies in mice showed that the nsP4-C483Y variant caused similar disease and viral tissue burden as WT CHIKV, while the nsP4-Q192L variant was strongly attenuated. Together these results support the potential of 4′-FIU to be an important antiviral against CHIKV.

## Author summary

The alphavirus chikungunya virus (CHIKV) is spread by mosquito vectors and currently causes millions of cases of human infections around the world. Initial symptoms of CHIKV infection include fever, rash, and severe joint and muscle pain. In a high proportion of cases infection progresses to chronic painful arthritis, and can even cause fatal encephalitis. There are no approved antiviral treatments for CHIKV infection. We recently demonstrated that the orally available compound 4′-fluorouridine (4′-FIU)

The sequence data are available from the NCBI, Bioproject accession number PRJNA1141983 in the NCBI BioProject database (https://www.ncbi.nlm.nih.gov/bioproject/), and GitHub (https://github.com/greninger-lab/Yin_CHIKV_fidelity_supp). Virus and reporter constructs are available upon request.

**Funding:** This work was supported by the National Institute of Allergy and Infectious Diseases U19 grant AI171403 Project 2 to MK, TEM, AM and ALG. The content of this paper is solely the responsibility of the authors and does not necessarily represent the official views of the NIH, NIAID, or NCI. The funders had no role in study design, data collection and interpretation, or the decision to submit the work for publication.

**Competing interests:** The authors have declared that no competing interests exist.

strongly inhibits CHIKV replication in cell culture and decreases viral disease symptoms in a mouse model of CHIKV infection. In order to advance 4′-FlU as a potential antiviral treatment, here we asked if 4′-FlU could select for CHIKV resistance or increased viral pathogenesis. We identified 2 mutations that decreased the sensitivity of CHIKV to inhibition by 4′-FlU. However, CHIKV variants containing those mutations were still inhibited by higher concentrations of 4′-FlU. In addition, the variants did not show increased virus growth and were not more pathogenic in mice. Together our data support the potential of 4′-FlU as an antiviral against CHIKV and related alphaviruses.

## Introduction

Chikungunya virus (CHIKV) is a mosquito-borne virus that is a global health concern. Acute CHIKV infection of humans can result in chikungunya fever, with symptoms including fever, rash, and severe muscle and joint pain [1,2]. In an estimated 51% of patients [3], this acute disease progresses to debilitating chronic polyarthritis [4]; infection can also result in fatal encephalitis [5–7]. CHIKV is a positive-strand enveloped RNA virus from the *Alphavirus* genus [8], which includes other important human pathogens such as Mayaro virus, Ross River virus, and the encephalitic Venezuelan, Western and Eastern equine encephalitis viruses [9]. Although the first CHIKV vaccine was recently approved for human use [10], there are as yet no licensed antivirals for alphavirus infection.

The ~12 kb CHIKV RNA genome has two open reading frames. The first encodes the four non-structural (ns) proteins (nsP1-nsP4), which are directly translated from the genomic RNA and mediate genome replication and transcription [11]. The second ORF encodes the virus structural proteins (C-E3-E2-6K/TF-E1), which are translated from a subgenomic RNA and assemble as progeny virions [11,12]. While antiviral strategies against CHIKV have considered both the structural and ns proteins as targets [13–17], the RNA replication complex is particularly promising given its critical enzymatic activities [11]. Alphavirus RNA replication occurs within membranous structures termed "spherules", which help to prevent the viral dsRNA intermediates formed during replication from triggering host cell immune responses [11,18,19]. Within the spherules, nsP1 anchors the replication complex by its association with the membrane and also caps the positive-strand nascent viral RNAs [20–22]. The N-terminal domain of nsP2 has RNA helicase and RNA triphosphatase activity required for RNA capping, and the C-terminal domain has protease activity critical for processing the ns polyprotein [23–26]. The most C-terminal portion of nsP2 displays structural similarity to RNA methyltransferases but appears to be enzymatically non-functional [27]. NsP3 contains a macrodomain, a central zinc-binding region, and a disordered region that is phosphorylated and recruits host factors [28–30]. Lastly, nsP4 is the alphavirus RNA-dependent RNA polymerase (RdRp) and mediates viral RNA synthesis [31].

The core structure of the alphavirus replication complex has been resolved using in vitro reconstitution and in situ electron cryotomography [32–34]. NsP1 forms a dodecameric ring structure anchored to the membrane at the neck of the spherules. NsP4 is positioned within the central pore of the nsP1 ring in a complex with the nsP2 helicase-protease. Activation of nsP4 occurs through its association with the nsP1 ring, and interaction with nsP2 further enhances the enzymatic activities of nsP4. During viral RNA replication, nsP4 synthesizes a negative sense RNA that forms a dsRNA intermediate with the genomic RNA and acts as a template for the synthesis of genomic and subgenomic RNAs. The nascent RNAs are capped by nsP1 and nsP2 after their transfer to the cytosolic surface of the replication complex. The

RNA is then exported to the cytosol for translation and for genomic RNA packaging into virions [32,33,35].

The alphavirus nsP4 is an important target for antivirals given its central importance in viral RNA synthesis and the fact that RdRp activity is absent from host cells. Prior studies have shown that ribavirin, a nucleoside analogue, blocks CHIKV replication, and inhibition of RdRp activity is one of the mechanisms [36]. Passaging of CHIKV in the presence of ribavirin or 5-fluorouracil selected for a C483Y substitution in nsP4, which has been characterized for effects on RdRp fidelity [37–39]. β-D-N4-Hydroxycytidine (NHC) is a nucleoside analogue that shows antiviral activity against CHIKV [40], SARS-CoV-2 [41,42], and influenza viruses [43] through generating deleterious mutations during viral replication. NHC treatment also inhibits replication of Venezuelan equine encephalitis virus (VEEV), and the VEEV nsP4 mutations P187S, A189V, and I190T confer resistance to NHC [44]. Favipiravir was originally designed as an inhibitor for influenza virus [45] but also can inhibit CHIKV replication [46], with resistance arising from a K291R mutation in nsP4 [47]. 4′-fluorouridine (4′-FlU, EIDD-2749) has shown highly potent inhibition of SARS-CoV-2, respiratory syncytial virus, and seasonal and highly pathogenic influenza viruses, with oral efficacy in animal models [48,49]. It mimics the physiological nucleotide uridine and inhibits RNA synthesis by triggering chain termination [50]. Recently, we demonstrated that 4′-FlU potently inhibits CHIKV RNA replication in vitro and reduces disease signs, viral tissue burden and inflammatory responses in mouse models of CHIKV and Mayaro virus infection [51].

In preparation for advancing 4′-FlU to clinical evaluation as a treatment for alphavirus infection, it is important to understand its potential to select for decreased virus drug sensitivity and/or more pathogenic phenotypes. Here we serially passaged CHIKV in cell culture in the presence of 4′-FlU. We identified CHIKV variants with reduced 4′-FlU sensitivity and demonstrated that decreased sensitivity was conferred by either Q192L or C483Y mutations in nsP4. These CHIKV variants did not have a detectable fitness advantage either in cell culture or in vivo. Studies in a CHIKV mouse model demonstrated that mice infected with the C483Y variant developed similar disease signs and viral tissue burden as mice infected with WT virus, while the Q192L mutation reduced both disease signs and viral burden. Our results support 4′-FlU as a candidate for an alphavirus antiviral therapy.

## Results

### 4′-FlU selects for resistance mutations in the CHIKV RNA replication complex

To characterize the resistance profile of 4′-FlU, we passaged an attenuated strain, CHIKV 181/25 [52], in U-2 OS, a human osteosarcoma cell line, in the presence of relatively low concentrations of 4′-FlU (Fig 1A). Cells were inoculated at an MOI of 1 focus forming unit (FFU)/cell; 6 independent wells were then cultured in the presence of inhibitor (4′-FlU lineage 1–6), and 3 wells in the presence of the DMSO vehicle (DMSO lineage 1–3) for 16 h (Fig 1B). Virus production after passage 1 (P1) was quantitated and the MOI was adjusted to 0.5 FFU/cell for the remaining rounds of selection. As predicted by our prior results [51], virus titers were reduced by 2–3 logs in the P1 4′-FlU selections compared with DMSO controls (Fig 1B). Inhibition of all 6 4′-FlU lineages was clearly decreased by P4 as compared to parallel drug treatments of an unselected 181/25 primary stock at the same MOI (Fig 1B). Two additional rounds of selection at a higher concentration of 4′-FlU were performed and the P6 stocks were then plaque-purified. One virus clone was plaque-purified and sequenced per lineage. RNA was isolated from the P6 stocks and the plaque-purified viruses, reverse-transcribed, and whole genome sequencing (WGS) of CHIKV genomes was performed by Illumina deep sequencing (Fig 1A).

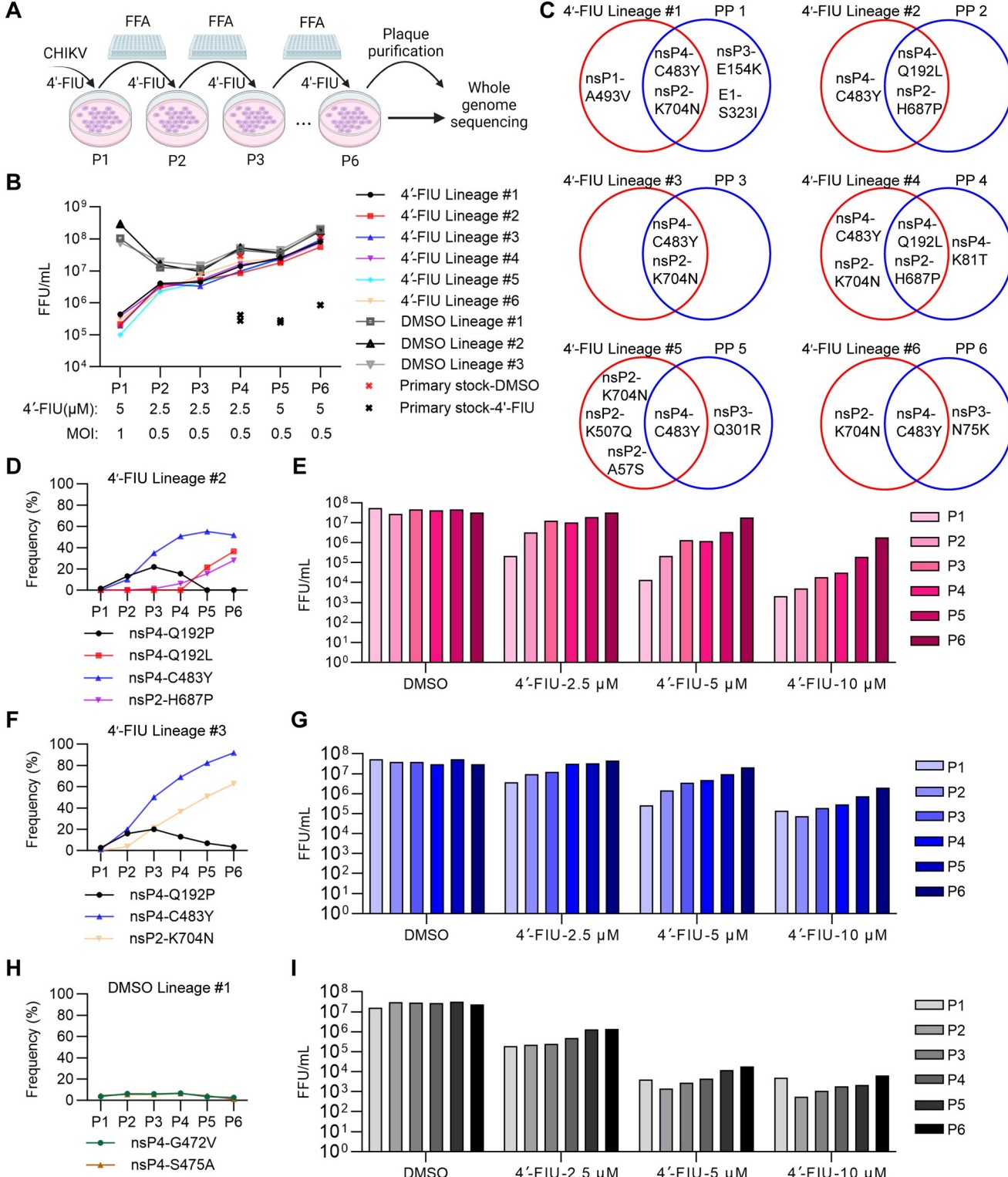

**Fig 1. 4′-FlU selects for mutations in the CHIKV RNA replication complex.** (**A**) Overall pipeline for in vitro selection of CHIKV by 4′-FlU. Fig 1A was prepared using BioRender. (**B**) Results of 6 passages of 6 independent virus stocks, termed lineage 1–6. Virus titer was quantitated by FFA after each passage in U-2 OS cells in the presence of the indicated concentration of 4′-FlU or DMSO vehicle. The MOI is indicated for each passage. Titers for primary (non-passaged) virus stocks are shown following parallel selection with 4′-FlU or DMSO vehicle. (**C**) Mutations identified in 6 virus lineages (red) and corresponding plaque-purified (PP) virus stocks (blue) by WGS. One virus clone was plaque purified and sequenced per lineage. (**D, F** and **H**)

Frequencies of mutations in passage 1 to 6 for 4′-FlU treated viral lineage #2 (**D**), lineage #3 (**F**) and DMSO treated lineage #1 (**H**), as determined by WGS. (**E, G** and **I**) Tests of virus production in the presence of the indicated concentrations of 4′-FlU or DMSO for passage 1 to 6 of viral lineage #2 (**E**), lineage #3 (**G**) and DMSO lineage #1 (**I**). U-2 OS cells were infected with virus from the indicated passages for 1 h at MOI = 1, and then cultured with the indicated concentrations of 4′-FlU for 24 h. Infectious virus production was quantitated by FFA.

Mutations in nsP4 and nsP2 were detected at high allele frequency in all of the 4′-FlU-selected P6 stocks (S1 Table). 4′-FlU lineages 1, 3, 5, and 6 contained an nsP4-C483Y mutation at an allele frequency from 83% to 93%, indicating that most viral genomes in these stocks contained this mutation. An nsP2-K704N mutation was also detected in these stocks at frequencies from 43% to 63%. 4′-FlU lineages 2 and 4 contained the nsP4-C483Y mutation plus an nsP4-Q192L mutation, with the combined frequency of these two nsP4 mutations at ~90%. An nsP2-H687P mutation was found with similar frequency as the nsP4-Q192L mutation. The DMSO lineages 1–3 did not contain any mutations with frequency above 11%. Plaque-purified viruses from 4′-FlU lineage 1 (PP1) and 4′-FlU lineage 3 (PP3) contained both the nsP4-C483Y mutation and the nsP2-K704N mutation (Fig 1C, S2 Table). PP5 and PP6 contained the nsP4-C483Y mutation, but rather than nsP2-K704N they contained either an nsP3-Q301R or nsP3-N75K mutation. Both PP2 and PP4 contained the nsP4-Q192L and the nsP2-H687P mutations. Thus, pairs of mutations in nsP4 and nsP2 were detected in both the 4′-FlU lineage stocks and plaque-purified viruses.

To follow the accumulation of mutations during passaging, P1 to P6 stocks from 4′-FlU lineages 2 and 3 and DMSO lineage 1 were analyzed by WGS and for inhibition by 4′-FlU. We detected a previously unobserved nsP4 mutation, Q192P, in early passages of lineages 2 and 3 (Fig 1D and 1F). This mutation was gradually outcompeted by the nsP4-C483Y mutation. In 4′-FlU lineage 2, the Q192L mutation in nsP4 also appeared at P5 and was likely competing with the C483Y mutation. The frequency of the corresponding nsP2 mutations increased with the nsP4 mutations during selection of 4′-FlU lineages 2 and 3. The resistance of both lineage 2 and 3 to 4′-FlU increased with the frequency of the paired nsP4-C483Y/nsP2-K704N or nsP4-Q192L/nsP2-H687P mutations (Fig 1E and 1G). Throughout passage of DMSO lineage 1, mutations were only detected with low frequency and did not significantly affect 4′-FlU sensitivity (Fig 1H and 1I).

## The nsP4-Q192L or nsP4-C483Y mutation is primarily responsible for reduced sensitivity to 4′-FlU

To define the role of the nsP4-C483Y/nsP2-K704N and nsP4-Q192L/nsP2-H687P mutations, we engineered them into the CHIKV 181/25 genome in pairs or separately and measured their effects on inhibition by 4′-FlU. The paired mutations resulted in reduced sensitivity to 4′-FlU compared with WT CHIKV, with ~3-4-fold increases in their $EC_{90}$ (Fig 2A and 2B). The single mutations nsP4-Q192L or nsP4-C483Y were primarily responsible for the decrease in sensitivity, with increases of ~3.4 or 3.9-fold in their $EC_{90}$, respectively. The nsP2 mutations did not cause significant increases in $EC_{90}$ either alone or in combination with the relevant nsP4 mutation (Fig 2B). We also tested the 4′-FlU sensitivity of an engineered nsP4-Q192L/C483Y double mutant. While the $EC_{90}$ of the double mutant was slightly higher than that of either nsP4 mutant alone, it also showed decreased viral titers (Fig 2A and 2B). We characterized the growth properties of the single and double nsP4 mutants by multi-step growth curves at low MOI in U-2 OS cells and C6/36 mosquito cells. Virus stocks containing the nsP4-Q192L/C483Y mutations showed reduced growth in both cell lines compared to WT or the single mutants, with the strongest reductions observed in mosquito cells (Fig 2C and 2D).

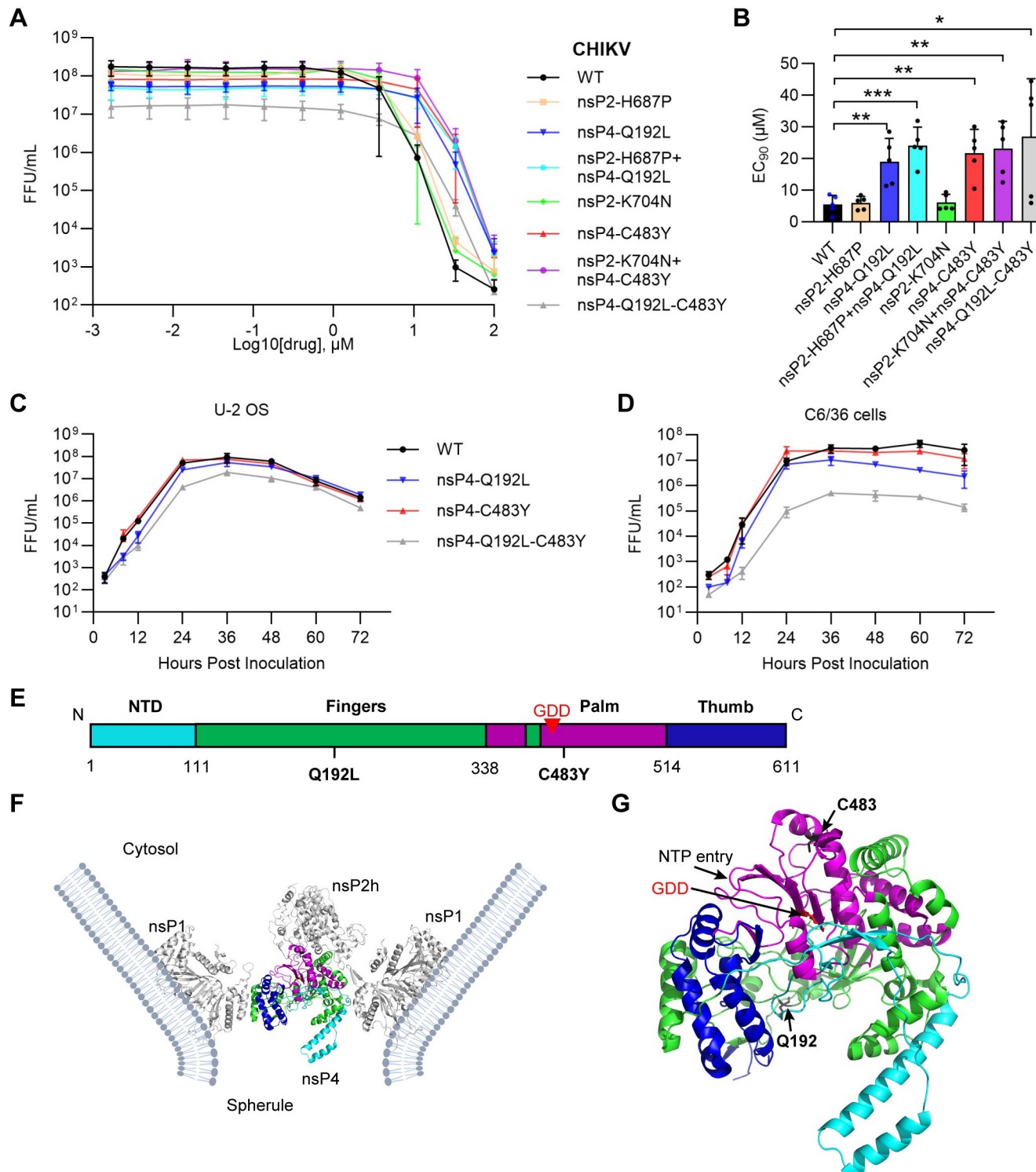

**Fig 2. nsP4-C483Y or Q192L mutations decrease sensitivity to 4′-FlU.** (**A**) Specific mutations were engineered into the CHIKV 181/25 genome, and the resultant virus stocks were tested as in Fig 1E for inhibition of virus production by the indicated concentrations of 4′-FlU. Data shown represent the mean ± s. d. of five independent experiments. (**B**) Mean $EC_{90}$ values ± s.d. determined from the data in panel A, with virus production normalized to that of the DMSO control. Individual data points are shown as dots. (**C** and **D**) Multi-step viral replication analysis of WT CHIKV and CHIKV with nsP4 mutations in (**C**) U-2 OS cells or (**D**) C6/36 cells. Cells were inoculated with the indicated virus (MOI = 0.1) for 2 h, washed, and virus production at the indicated times (3, 8, 12, 24, 36, 48, 60, 72 hpi) quantitated by FFA. The graphs represent the means of 2 independent experiments. (**E**) Linear diagram of the CHIKV nsP4 sequence, indicating the N-terminal domain (NTD) in cyan, and the RdRp finger region in green, palm in purple and thumb in blue. The positions of Q192, C483 and

the conserved GDD active-site residues are marked. (**F**) Structure of the viral replication complex (CHIKV-nsp1+CHIKV-nsP2+ ONNV-nsP4) at the neck of the membrane spherule, with nsP1 forming a dodecameric ring that interacts with the membrane, and nsP4 and the nsP2 helicase domain (nsP2h) within the central pore of the ring. PDB: 7Y38 [33] (**G**) The structure of nsP4, colored as in (**E**), with the GDD residues indicated by the arrow, and Q192 and C483 labeled and shown in black sticks. The position of the NTP entry site at the nsP4 palm domain is indicated. Statistical significance in B was calculated by unpaired two-tailed t-tests. *P < 0.05, **P < 0.01, ***P < 0.001. Mean $EC_{90}$ values for nsP4-Q192L vs nsP2_H687P+nsP4-Q192 or nsP4-C483Y vs nsP2-K704N+nsP4-C483Y did not show significant differences.

The two nsP4 mutations are in residues that are conserved across the *Alphavirus* genus and in CHIKV strains including recent clinical isolates (S1A, S1B, S2A and S2B Figs), with Q192L located in the RdRp fingers domain and C483Y in the palm domain (Fig 2E and 2G). Previous studies showed that C483Y reduced the sensitivity of CHIKV to inhibition by ribavirin [37,38], and thus this mutation is not specific to 4′-FlU selection. The prior C483Y study also identified a G641D nsP2 mutation, which was shown to act synergistically with nsP4-C483Y to increase resistance to ribavirin [39]. The two nsP2 mutations we identified are in residues not conserved among alphaviruses but are conserved in CHIKV strains including recent clinical isolates (S3 and S4 Figs). Both of these mutations and the previously identified nsP2 G641D are located in the nsP2 methyltransferase-like (MTL) domain (S5 Fig), which is likely involved in interaction with viral RNA. The MTL domain is not visualized in the current structure of the replication complex [33], and thus the molecular features of potential crosstalk between our paired nsP4 and nsP2 mutations are unknown (Fig 2F).

## The Q192L and C483Y mutations decrease the 4′-FlU sensitivity of CHIKV RdRp

We next tested the effects of the nsP4 mutations on CHIKV RdRp activity. Cells were infected with WT CHIKV or the nsP4-Q192L and C483Y mutants, incubated in the presence or absence of 10 μM 4′-FlU, and stained to detect expression of nsP4 and the dsRNA replication intermediate (Fig 3A). Both WT and mutant-infected cells were positive for nsP4 and dsRNA in the absence of inhibitor, while only the mutant-infected cells were positive after treatment with 4′-FlU. We then used a CHIKV trans-replicase assay to further characterize the mutants (Fig 3B). In this assay, the nsP1-3 polyprotein and nsP4 are co-expressed with a reporter RNA that can be used as a template by the nsP4 RdRp, thus dissociating nsP4 enzymatic activity from nsP4 expression [53,54]. Amplification of the template RNA represents viral RNA synthesis and is detected by increased Firefly luciferase (Fluc) expression, while the RNA synthesized from the SG promoter is detected by *Gaussia* luciferase (Gluc) expression (Fig 3B). Results demonstrated that both the nsP4-Q192L and C483Y mutations decreased the 4′-FlU sensitivity of RNA synthesis (Fig 3C and 3D) in this in vitro assay. Together these data indicate that the Q192L and C483Y mutations act on CHIKV RdRp activity to decrease virus sensitivity to inhibition by 4′-FlU.

## The nsP4-C483Y and nsP4-Q192L mutations do not confer a virus fitness advantage

To compare the fitness of WT and the nsP4 mutants in *vitro*, U-2 OS cells or C6/36 cells were inoculated at low MOI with mixtures of the indicated CHIKV 181/25 WT or mutants for 24 h. The resultant virus stocks were serially passaged on new cells two more times and the frequency of mutations at each passage was determined by WGS (Figs 4 and S6). The nsP4-Q192L mutant was quickly outcompeted by WT in both cell lines and fitness was not rescued by inclusion of the nsP2-H687P mutation (Fig 4A–4C, 4E–4G). Competition studies between the two mutants showed that nsP4-C483Y had a fitness advantage over nsP4-Q192L

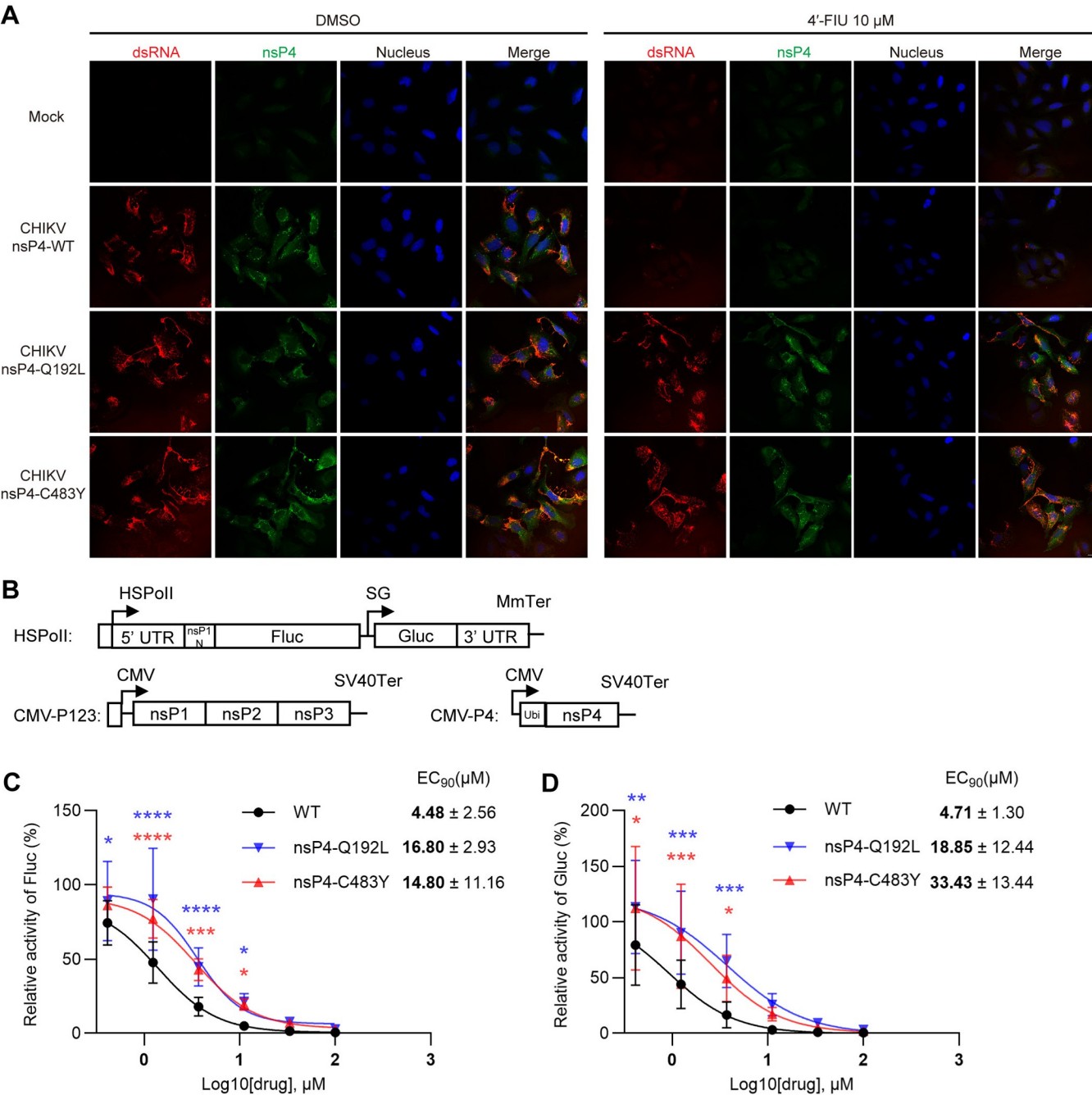

**Fig 3. The nsP4-Q192L and C483Y mutations decrease the 4′-FIU sensitivity of both replication and transcription.** (**A**) Effect of the mutations on synthesis of dsRNA in the presence of 4′-FIU. U-2 OS cells were infected with the indicated CHIKV 181/25 variants at MOI = 3 for 1 h, incubated with 10 μM 4′-FIU, fixed and permeabilized at 8 hpi, and stained with antibodies against dsRNA (red), nsP4 (green), and Hoechst (nucleus). The figure shows representative images from two independent experiments. Scale bar = 20 μm. (**B**) Diagrammatic presentation of expression plasmids for the CHIKV trans-replicase assay including template RNA (top) and replicase protein expression constructs. HSPolI, a truncated promoter (residues 211 to −1) for human RNA polymerase; MmTer, a terminator for RNA polymerase I in mice; CMV, immediate early promoter of human cytomegalovirus; SV40Ter, simian virus 40 late polyadenylation region. (**C** and **D**) 4′-FIU sensitivity of the CHIKV replicase harboring nsP4 mutations. U-2 OS cells were transfected with CHIKV replicase constructs for 4 h and the indicated concentrations of 4′-FIU were then added. Fluc and Gluc activities were measured 20 h after transfection, and normalized to those of DMSO-treated controls. Data shown represent the mean ±s.d. from three independent experiments. The mean $EC_{90}$ values ± s.d. are shown to the right. Statistical significance was determined by two-way ANOVA with Dunnett's multiple comparison test compared to WT. *$P < 0.05$, **$P < 0.01$, ***$P < 0.001$, ****$P < 0.0001$. Note that the CHIKV label in panel A is used to differentiate results with virus from those in panel B with the trans-replicase system.

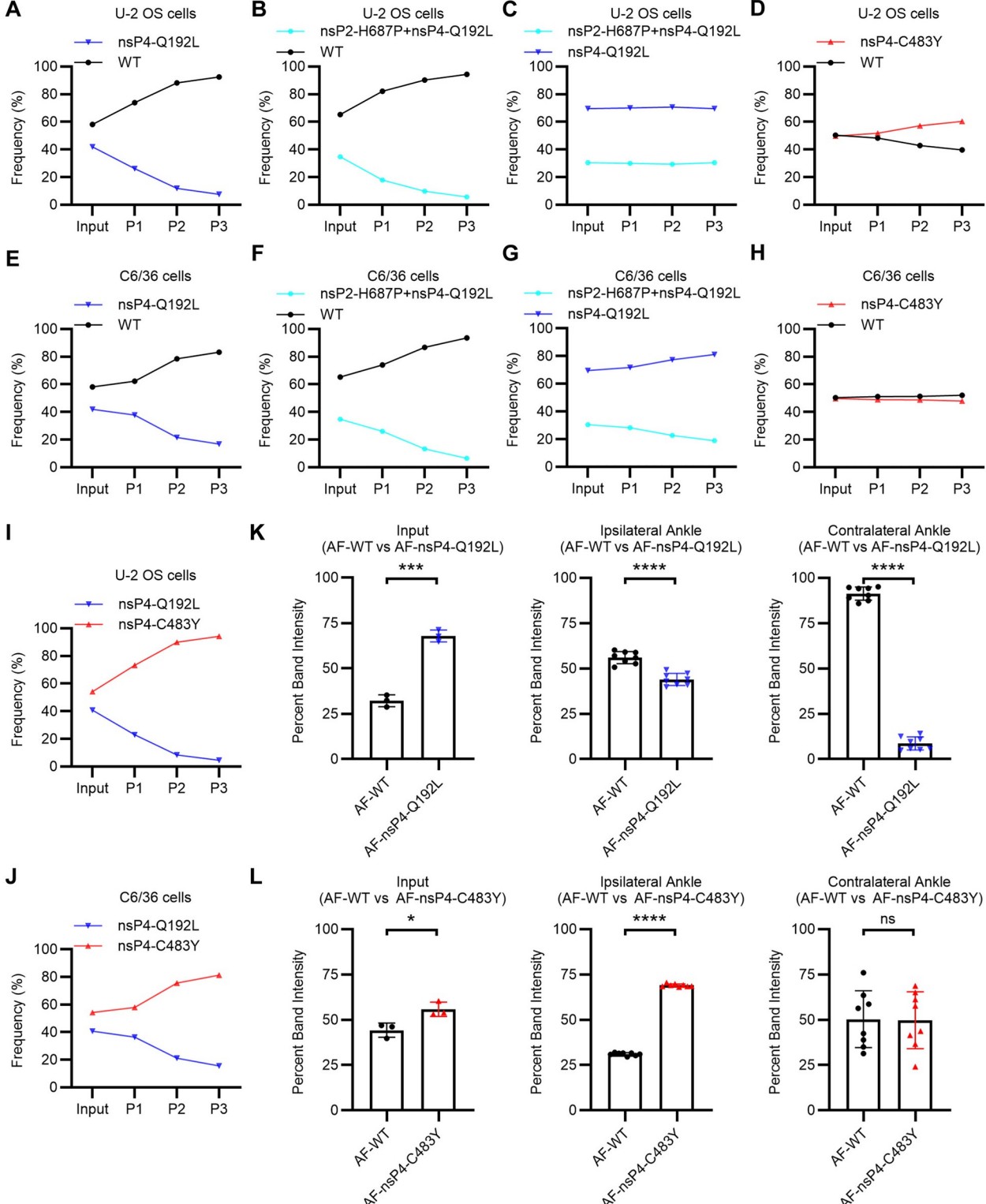

**Fig 4. The nsP4-C483Y and nsP4-Q192L mutations do not provide an in vitro or in vivo fitness advantage. (A-D, I)** Competition assays comparing relative viral fitness in U-2 OS cells. Cells were inoculated at an MOI of 0.1 FFU/cell with 1:1 mixtures of the indicated 181/25 viruses and cultured for 24 h. Viral titers were quantitated by FFA, and the viruses were serially passaged on new cells two more times at MOI of 0.1 FFU/cell. RNA was extracted from the input and P1-P3 passages, cDNA was generated, and viral frequencies were determined by WGS. The % frequencies represent the averages of three technical replicates from one experiment with each condition. **(E-H, J)** Competition assays comparing

relative viral fitness in C6/36 cells. Cells were inoculated and viruses serially passaged and analyzed as in panels A-C. The % frequencies represent the averages of three technical replicates from one experiment with each condition. Panels B and F report the frequency of nsP4-Q192L to represent the nsP2-H687P+nsP4-Q192L mutant. Panels C and G report the frequency of nsP2-H687P to distinguish the two competitors. (**K-L**) Competition assays comparing relative viral fitness in vivo. C57BL/6 mice were inoculated in the left footpad with 1,000 FFU of 1:1 mixtures of CHIKV AF15561 WT (marked with an ApaI restriction site) and the indicated nsP4 mutant virus. At 24 h post-inoculation, ipsilateral and contralateral ankle tissues were collected, RNA was extracted, and cDNA was generated and PCR amplified. Digestion of the PCR product was used to determine the ratios of ApaI marked to unmarked virus, and the percent band intensity is displayed. Mean ±s.d. N = 8, two experiments. Statistical significance in G-H was calculated by Student's unpaired two-tailed t-tests. *P < 0.05, ***P < 0.001, ****P < 0.0001, ns not significant.

in both cell lines (Fig 4I and 4J). The frequency of the nsP4-C483Y mutation was slightly increased vs. WT in U-2 OS cells after 3 passages, but was approximately equivalent in C6/36 cells (Fig 4D and 4H). Fitness of the C483Y mutant vs. WT was not significantly altered by inclusion of the nsP2-K704N mutation, which showed a small competitive advantage vs. C483Y alone (S6 Fig).

To test the fitness of the mutants in vivo, the nsP4-C483Y and Q192L mutations were engineered into CHIKV strain AF15561 [52], a pathogenic clinical isolate that is the parent virus to CHIKV 181/25. For competition experiments the WT CHIKV AF15561 was genetically marked with an ApaI restriction site (AF-WT) [55]. Fitness was assessed by competing each mutant virus against the marked WT CHIKV AF1556 and differentiating the two competitors based on digestion with the ApaI restriction enzyme (see also methods section). C57BL/6 mice were inoculated in the left footpad with mixtures of marked AF-WT and unmarked AF15561 containing the nsP4 mutations. Ipsilateral and contralateral ankle tissues were harvested at 24 h post-inoculation and the ratios of AF-WT to mutant viruses were measured. The Q192L mutant was strongly outcompeted by AF-WT, as evidenced especially in the contralateral ankle tissue which reflects dissemination and secondary infection of a distal site (Fig 4K). The C483Y mutant showed equivalent infection to the AF-WT in contralateral ankle tissue (Fig 4L). Collectively, these findings suggest that C483Y does not detectably alter viral fitness, while Q192L confers a fitness cost.

### The nsP4-C483Y and nsP4-Q192L mutations do not change nsP4 fidelity

The nsP4-C483Y mutation was previously identified by selecting CHIKV for resistance to ribavirin or 5-fluorouracil and was shown to confer resistance to both of these compounds [37]. While the mutation was originally found to increase replication fidelity [37], a more recent study reported that C483Y caused no significant difference in fidelity [56]. To investigate the effect of the CHIKV nsP4-C483Y and Q192L mutations on RdRp fidelity, we infected U-2 OS and C6/36 cells with P0 stocks of WT and the nsP4-C483Y and Q192L mutants prepared from the respective 181/25 infectious clones. The sequence diversity after a single 24-h passage (P1) was evaluated by WGS of the P0 and passaged populations. There were no statistically significant differences in the mutant and WT mutation frequencies across the complete viral genomes (Fig 5A). We checked for differences in the number of mutated sites, defined as any nucleotide position showing evidence of a substitution regardless of frequency. No differences in the number of mutated sites were observed between CHIKV C483Y and Q192L vs. the WT (Fig 5B). We also evaluated overall population diversity by Shannon entropy, which is suitable for comparing low-frequency variants between groups (see methods for details). As shown in Fig 5C, Shannon entropy was not significantly different across CHIKV variants and WT. To compare mutations with different frequencies, we analyzed the population diversity for the number of single nucleotide polymorphisms (SNPs). The nsP4-C483Y and Q192L populations did not show significant differences in SNPs vs. the WT, and no SNPs with a

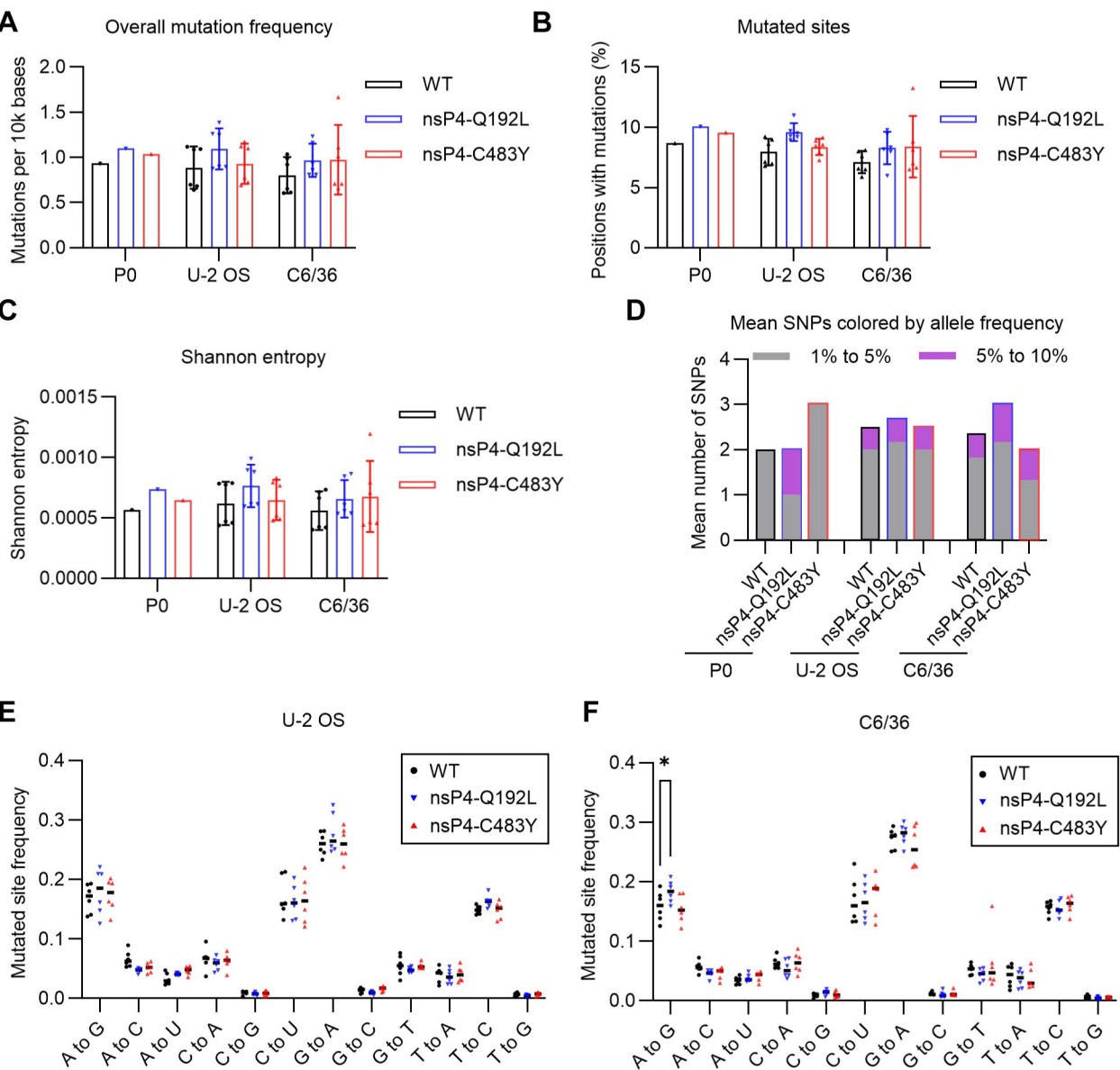

**Fig 5. The nsP4-C483Y and nsP4-Q192L mutations do not change the fidelity of nsP4.** P0 virus stocks were prepared from 181/25 WT or mutant infectious clones, and used to inoculate U-2 OS or C6/36 cells in triplicate at an MOI of 0.1 FFU/cell. Virus was harvested 24 h later and sequenced by WGS. Two independent experiments, each with three technical replicates, were performed. (**A and B**) Mutation frequencies (**A**) and sites (**B**) for WT, Q192L and C483Y measured by WGS across the whole genome. The overall mutation frequency was calculated as the number of mutated nucleotides per 10,000 mapped nucleotides after quality and depth filtering. A mutated site was defined as any nucleotide position with evidence of a substitution regardless of frequency. (**C**) Diversity of CHIKV populations as measured by Shannon entropy. (**D**) Mean number of SNPs across CHIKV genome. No SNPs were observed at more than 10% frequency in CHIKV C483Y, Q192L and WT. (**E and F**) Mutational spectra of CHIKV populations by mutated site frequency after growth in (**E**) U-2 OS and (**F**) C6/36 cells. Statistical significance was determined by two-way ANOVA compared to WT with Dunnett's multiple comparison test (E and F). *P < 0.05.

frequency of more than 10% were observed (Fig 5D). There were no nucleotide substitutions that were significantly enriched in the mutant vs. the WT virus populations save for a small increase in A to G transitions for the Q192L mutant in C6/36 cells (P = 0.045) but not in U-2 OS cells (Fig 5E and 5F). Together these results indicate that the CHIKV nsP4-Q192L and C483Y mutations did not produce significant differences in replication fidelity.

## Pathogenesis of the CHIKV nsP4-C483Y and nsP4-Q192L mutants in vivo

We first confirmed the effect of the nsP4-C483Y and Q192L mutants on the pathogenic CHIKV strain AF15561 in normal human dermal fibroblasts treated with 4′-FlU. Similar to our previous results with CHIKV 181/25, the mutations decreased AF15561 sensitivity to 4′-FlU and increased the $EC_{90}$ relative to WT (Fig 6A). To evaluate the effect of the mutations on CHIKV pathogenesis, 4-week-old WT C57BL/6 mice were inoculated in the left rear foot pad with $10^3$ PFU WT or mutant virus. Joint (ankle) swelling and body weight were measured daily as markers of disease (Fig 6B). Mice inoculated with WT CHIKV displayed joint swelling and reduced weight gain. Mice inoculated with the Q192L mutant showed significantly less joint swelling while mice inoculated with the C483Y mutant exhibited increased joint swelling, although the increase was below the level of significance (Fig 6C). Infection with either WT or the C483Y mutant led to significantly reduced weight gain, while the weight gain in mice inoculated with Q192L was similar to that of uninfected control mice (Fig 6D). Virus tissue burden was determined by RT-qPCR at 7 days post-inoculation (dpi). The viral burden of the Q192L mutant was significantly reduced in the ipsilateral ankle, contralateral ankle and spleen, while the C483Y mutant produced similar tissue viral burdens as those of the WT virus (Fig 6E). These results demonstrate that infection by the C483Y mutant did not increase viral burden compared to WT virus in any of the sampled tissues, in agreement with prior studies of CHIKV C483Y [56]. Our in vivo characterization of the novel nsP4 mutant Q192L demonstrated that the mutation reduced both CHIKV disease signs and viral burden.

## Discussion

Our prior studies demonstrated that 4′-FlU is a potent inhibitor of CHIKV replication in vitro, and significantly reduces disease and virus tissue burden in mice [51]. Given the lack of proof-reading by the alphavirus RdRp, it was anticipated that CHIKV could acquire mutations that would decrease its sensitivity to 4′-FlU, and potentially increase virus replication and pathogenesis. A recent study selected for 4′-FlU resistance of influenza A virus and reported three distinct clusters of RdRp mutations that decrease virus inhibition by 4′-FlU [57]. Notably, however, these mutations impose a substantial fitness cost as the variants are attenuated in both cell culture and in vivo infection models. In this study, we identified two sets of mutations, nsP2-H687P/nsP4-Q192L and nsP2-K704N/nsP4-C483Y, which occurred in multiple independent selections with 4′-FlU. Using reverse genetics, we demonstrated that the single mutations in nsP4, Q192L and C483Y, were primarily responsible for drug resistance, although the paired nsP2 mutations gave small additional increases. Trans-replicase assays confirmed the specific effect of the Q192L and C483Y mutations on the 4′-FlU sensitivity of nsP4.

In the structure of the reconstituted core CHIKV replication complex with ONNV nsP4 [33], Q192 is located in the index subdomain of the fingers domain. This region is involved in conformational changes that maintain the active closed structure of the catalytic site during polymerization. Since the reconstituted core structure of the CHIKV replication complex does not contain viral RNA, it is unclear if Q192 contributes directly to RNA binding. In the structure of the RRV nsP4 [31], the hydrophilic Q192 was replaced with alanine to improve crystal diffraction, and this residue is believed to be exposed on the surface of the RdRp protein. The large number of residues in the fingers domain that were not visualized in the structure is in keeping with this region being highly flexible and dynamic.

C483 is highly conserved and lies in the nsP4 palm domain. The palm domain contains five polymerase motifs, A-E, with C483 located in a helix between the active site GDD residues in motif C and motif D. When compared with the known structures of positive-strand virus

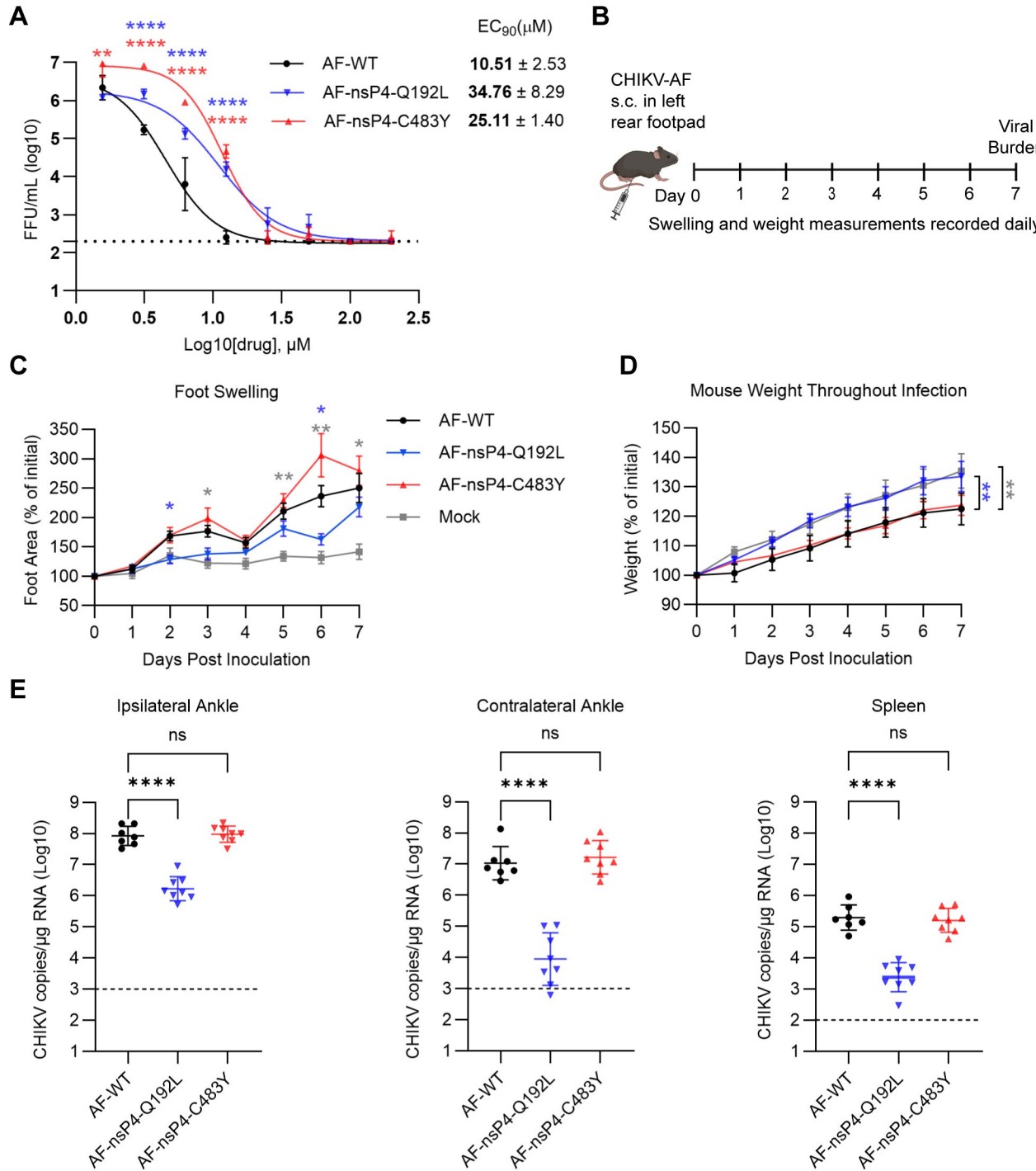

**Fig 6. Pathogenesis of CHIKV nsP4-C483Y and nsP4-Q192L.** (**A**) Dose response assays of CHIKV AF15561 strain containing the indicated mutations. Normal human dermal fibroblasts were inoculated with the indicated CHIKV (MOI = 1) for 1 h, and then cultured with the indicated concentrations of 4′-FlU for 24 h. Virus production was quantitated by FFA. Data shown represent the mean ± s.d. from three independent experiments. Mean EC$_{90}$ values ± s.d. are shown. (**B**) Schematic of experimental design. WT C57BL/6 mice (n = 8 mice/group) were inoculated with $10^3$ PFU of CHIKV in the left rear footpad. Fig 6B was prepared using BioRender. (**C** and **D**) Swelling of the ipsilateral foot and quantitation of mouse weight over time, measured daily. (**E**) At 7 dpi, the viral burden in the tissues indicated was quantified by RT-qPCR. Statistical significance was determined by two-way ANOVA compared to WT with Dunnett's multiple comparison test (A), with Tukey's multiple comparison test (C), repeated one-way ANOVA with Tukey's multiple comparison test (D) or one-way ANOVA with Dunnett's multiple comparison test (E). *P < 0.05, **P < 0.01, ****P < 0.0001, ns not significant.

RdRps [31], motifs A, B, C and E are structurally well-conserved and constitute the core palm domain, while motif D is not conserved. The residues in motif D form an elongated β-strand conformation, which differs from the canonical β-strand-loop motif and distinguishes alphaviruses from other viral RdRps [31]. The structure of alphavirus motif D may confer decreased flexibility, allowing this elongated motif to inhibit the entry of nucleoside triphosphate NTPs in the apoenzyme state [58]. Considering the conservation of C483 among alphaviruses, it could be interesting to determine if the C483Y mutation affects motif D flexibility and the entry of NTPs into nsP4.

Prior studies of the effect of the CHIKV nsP4-C483Y mutation on replication fidelity have yielded variable results. This CHIKV mutation was initially identified by selecting for virus resistance to mutagenesis by ribavirin and 5-fluorouracil [37]. An increase in mutant replication fidelity was observed by evaluation of the partial E1 sequences of independent virus stocks prepared by genome electroporation, and by deep sequencing of fragmented 1 kb amplicons [37]. However, a follow-up paper showed only a marginal fidelity difference under normal culture conditions, which was magnified by ribavirin treatment [39]. A later study using random-primed WGS methods did not find that the nsP4-C483Y mutation increased fidelity [56]. Here we evaluated fidelity by passage of WT and mutant virus on U-2 OS or C6/36 cells and WGS sequence comparison of the P0 and P1 stocks. No changes in mutation frequency or nucleotide substitution across the genome were detected. It was also reported that replacement of a portion of the SFV nsP2 region with that of CHIKV selects for the nsP4-C483Y mutation [59]. Given its occurrence under these varied conditions, nsP4-C483Y may represent a replicase hotspot that is selected in response to different stresses. We found that the nsP4-C483Y mutation did not affect CHIKV growth in U-2 OS or C6/36 cells or in our in vivo mouse model. Reduction of weight gain was similar to that observed in WT-infected mice.

NsP4-Q192L is a 4′-FlU-selected mutation that to our knowledge has not been found in selections with other alphavirus inhibitors. The mutation had no detectable effect on replication fidelity. Virus growth was not significantly impaired in either U-2 OS or C6/36 cells, but the mutant was outcompeted by CHIKV WT in vitro and in vivo. In addition, studies in mice showed that nsP4-Q192L was significantly attenuated in joint swelling, weight reduction, and viral burden. During the selection process, an nsP4 Q192P mutation arose first, but then appeared to be out competed by either the nsP4-C483Y or Q192L mutations. Selection for NHC resistance of VEEV identified 3 key nsP4 mutations in close proximity to Q192: P187S, A189V, and I190T [44]. While NHC acts as a mutagen and 4′-FlU has been defined as a chain terminator [50], the similar location of their resistance mutations suggests that this location may be involved in nucleotide recognition or incorporation.

In summary, serial passaging of CHIKV in the presence of 4′-FlU identified independent virus isolates with decreased drug sensitivity. Two mutations in nsP4, Q192L or C483Y, were demonstrated to be primarily responsible for the decrease in 4′-FlU sensitivity. CHIKV variants containing these mutations were still inhibited by higher concentrations of 4′-FlU. The mutations did not provide a fitness advantage in vitro or in vivo, and CHIKV pathogenesis in our mouse model was either unaltered or decreased. Thus 4′-FlU has the potential to be important in future CHIKV therapeutic strategies.

## Materials and methods

### Ethics statement

This study was performed in accordance with the recommendations in the Guide for the Care and Use of Laboratory Animals of the National Institutes of Health. Animal studies were conducted following approved institutional animal care and use committee (IACUC) protocols

(#00026) of the University of Colorado School of Medicine (Assurance Number A3269-01). Experimental animals were humanely euthanized at defined endpoints by exposure to isoflurane vapors followed by bilateral thoracotomy.

## Cell lines

U-2 OS cells (ATCC HTB-96) were cultured in modified McCoy's 5A medium supplemented with 10% fetal bovine serum (FBS). BHK-21 cells (ATCC CCL-10) were grown in high glucose Dulbecco's modified Eagle's medium (DMEM, Cytiva) supplemented with 10% FBS, 10% tryptose phosphate broth, and 0.29 mg/mL L-glutamine. C6/36 cells were cultured in DMEM (Cytiva) with 10% heat inactivated FBS. Growth media all contained 100 U penicillin/mL and 100 μg streptomycin/mL. Primary normal human dermal fibroblasts (ATCC PCS-201-012) were cultured in high glucose DMEM (Cytiva) with 10% FBS, 100 U penicillin/mL and 100 μg streptomycin/mL. U-2 OS, BHK-21 cells and normal human dermal fibroblasts were grown at 37˚C in 5% $CO_2$ atmosphere. C6/36 cells were cultured at 28˚C in 5% $CO_2$ atmosphere.

## Viruses

Virus stocks were produced from the following infectious cDNA clones: CHIKV 181/25 (pSin-Rep5-181/25ic [60], provided by Dr. Terence S. Dermody), CHIKV AF15561 [60], a virulent Southeast Asian human isolate that is the parent strain to 181/25, and the indicated mutants of 181/25 and AF15561 generated as described below. Infectious cDNA clones were linearized and purified, and capped RNA transcripts were generated using an SP6 in vitro transcription kit (Life Technologies) and electroporated into low-passage BHK-21 cells. Cells were cultured at 37˚C for 27–30 h until cytopathic effects were observed. Culture supernatants were then collected and clarified by centrifugation at 3,000 rpm in the Thermo Scientific TX-1000 rotor for 20 minutes at 4˚C. Virus stocks used in vitro assays were titered by focus formation assays (FFA) in U-2 OS cells.

## Site directed mutagenesis using Gibson assembly

To generate mutations in the infectious cDNA clones of CHIKV 181/25 and AF15561 using Gibson assembly, polymerase chain reaction (PCR) amplification of fragments containing mutations were performed using multiple primers, as detailed in S3 Table. The PCR fragments were then inserted into the 181/25 or AF15561 infectious clones linearized by digestion with Age1 and Swa1, using the Gibson assembly kit according to the manufacturer's instructions (#E2611, New England Biolabs).

## Focus formation assay

U-2 OS cells were seeded at $1.5 \times 10^4$ cells/well in 96 well-plates and cultured for 24 h. Cells were infected with 10-fold serial dilutions of CHIKV in Med-A (Minimum essential medium plus 0.2% BSA and 10 mM HEPES) for 2 h then overlaid with 1% carboxymethylcellulose in modified Eagle's Medium supplemented with 2% FBS and 10 mM HEPES pH 7.4. At 18 hpi, cells were fixed by adding 100 μl warm 1% paraformaldehyde (Electron Microscopy Science) in PBS to the overlay and incubating for 1 h. After 5 washes with PBS, the cells were permeabilized with 0.1% saponin in PBS containing 0.1% BSA. Cells were stained with mAb to E2 (E2-1) using hybridoma supernatants at a 1:10 dilution [61] followed by treatment with horseradish peroxidase-conjugated goat anti-mouse antibody. Foci were developed using TrueBlue Peroxidase substrate (#5510–0030, Seracare) and quantitated using an ImmunoSpot S6 Macroanalyzer (Cellular Technologies).

## Resistance selection

U-2 OS cells were seeded at $2 \times 10^5$ cells/well in 6-well plates, cultured for ~24 h, and inoculated with CHIKV 181/25 primary stock at an MOI of 1 FFU/cell. After 2 h, media containing 5 μM 4′-FlU was added to 6 wells, DMSO vehicle alone was added to 3 wells, and cultures were incubated for 16 h (P1). Virus stocks were frozen between passages, virus production was measured by FFA, and the MOI was adjusted to 0.5 FFU/cell for the remaining 5 rounds of selection (P2-P6) at the 4′-FlU concentration indicated in Fig 1B. At selection rounds 4, 5, and 6, controls were performed using additional wells of U-2 OS cells infected with unselected primary stock and treated in parallel with 4′-FlU or DMSO. Viruses were plaque-purified from the P6 stocks and amplified in culture. RNAs were isolated from virus stocks and plaque-purified samples, reverse-transcribed, and sequenced by WGS.

## CHIKV whole genome sequencing (WGS)

CHIKV isolate genomes were sequenced using metagenomic WGS, as described previously [62]. Briefly, viral RNA was reverse transcribed using random hexamers and Superscript IV (Thermo Fisher), followed by double-stranded cDNA synthesis using Sequenase v2.0 (Thermo Fisher) and library tagmentation using the Illumina DNA Prep (S) kit and 14 cycles of dual-indexed PCR. Libraries were sequenced 2x150bp on the NextSeq 2000.

For the initial resistance profiling experiments (Fig 1C–1H), and competition assays (Fig 4A–4F), paired sequencing reads were filtered for length and quality using fastp [63] with ≥60% of bases having a phred score ≥30 and a minimum length of 75 bases. Variant analysis was performed using RAVA (https://github.com/greninger-lab/RAVA_Pipeline) and the CHIKV 181/25 reference (Genbank MW473668.1).

For the fidelity analysis (Fig 5), paired sequencing reads were filtered using fastp [63] with minimum phred score of 30 per base, minimum overlap of 50 bases per read pair, and 100% agreement between those overlapping bases. The filtered and merged paired-end reads were downsampled using seqtk so each specimen had approximately equal sequencing depth: 400,000 reads per specimen and minimum coverage depth of 300 at each base. Consensus genomes for the P0 stocks were generated using Geneious 2023.1.2. The reads were aligned to the corresponding consensus genome from the P0 stocks using bwa-mem [64] with parameters from RAVA and variants were extracted using VarScan2 readcounts [65]. Custom scripts were used to parse the vcf files and perform fidelity analysis on the coding sequence, available on github at (https://github.com/greninger-lab/Yin_CHIKV_fidelity_supp). Mutation frequencies were calculated as follows: number of mutated bases per 10,000 mapped bases (mutations per 10k bases) [66]. Similar to work from Coffey et al. [37], the number of nucleotide positions in the CDS with a mutation, regardless of allele frequency, was also measured (i.e. mutated sites) [56]. Sequencing data can be found at BioProject PRJNA1141983. Shannon entropy is a commonly used metric of diversity (or information). It was calculated using the diversity package in R from a matrix of the counts of A, C, T, G's at each locus across the CHIKV reference genome from deep sequencing data for each condition and replicate. If a polymerase mutation were more associated with error, it would give a higher Shannon entropy value.

## Dose response assays and growth curves

$7.5 \times 10^4$ U-2 OS cells/well were seeded into 24-well plates and cultured for 24 h. The cells were infected with the indicated CHIKV strains at MOI = 0.1 in Med-A (MEM plus 0.2% BSA and 10 mM HEPES pH 7.0) for 1 h and washed 3 times with complete medium. Cells were then incubated in complete medium containing 3-fold serial dilutions of 4′-FlU. At 24 hpi, culture

supernatants were clarified by centrifugation and virus quantitated by FFA. Relative virus production was normalized to that of DMSO controls. Dose response curves were further analyzed by log(agonist) vs. response-Find ECanything to determine 90% $EC_{90}$ with Prism 10 (GraphPad).

For multi-step growth curves, $7.5 \times 10^4$ U-2 OS or C6/36 cells/well were seeded into 24-well plates and cultured for 24 h. The cells were infected with the indicated CHIKV strains at MOI = 0.01 in Med-A for 2 h, washed 3 times with complete medium and cultured in complete medium in the absence of drug. Culture supernatants were collected at the indicated hpi, clarified by centrifugation, and virus quantitated by FFA.

## Immunofluorescence microscopy

$1.5 \times 10^5$ U-2 OS cells/well were seeded onto glass coverslips in six-well plates, cultured for ~24 h, infected with the indicated CHIKV strains at MOI = 3 in Med-A for 1 h, washed 3 times with complete medium and treated with 10 μM 4′-FlU at 1 hpi. Cells were fixed at 8 hpi, permeabilized and stained as previously described [67], using primary antibody to dsRNA (mAb J2, Scicons, J2-1406, 1:300) and rabbit polyclonal antiserum to nsP4 (in-house). Confocal images were acquired using 63x oil immersion lens and a Leica TCS SP8 microscope in the Einstein Analytical Imaging Facility.

## *Trans*-replication assays

Approximately $3 \times 10^4$ U-2 OS cells were plated per well in 48-well plates, cultured for 24 h, then co-transfected with 250 ng plasmid encoding template RNA, 250 ng of plasmid encoding P123 WT, and 185 ng of plasmid encoding WT nsP4 or its variants harboring C483Y or Q192L mutations. Transfections were performed using LipoFectamine LTX with PLUS reagent (ThermoFisher Scientific) according to the manufacturer's instructions. At 4 h post-transfection, the media were replaced with fresh media containing 4′-FlU at concentrations of 100 μM, 33.3 μM, 11.1 μM, 3.7 μM, 1.2 μM or 0.4 μM, or DMSO alone (vehicle control). Cells were incubated for 20 h at 37°C, lysed and Fluc and Gluc activities were measured using the Dual-Luciferase Reporter Assay System (Promega). Fluc and Gluc activities were normalized to those of DMSO-treated controls. Dose response curves were analyzed by log(agonist) vs. response -Find ECanything to determine $EC_{90}$ with Prism 10 (GraphPad).

## Fidelity assay

$1 \times 10^5$ U-2 OS or C6/36 cells/well were seeded in 24-well plates, cultured for 24 h, then infected in triplicate at an MOI of 0.1 FFU/cell with P0 stocks of the indicated CHIKV variants prepared from the infectious clones. Culture supernatants (P1) were harvested at 24 hpi, viral RNA was extracted from the clarified supernatants using the RNEasy mini kit (QIAGEN, #74104) and eluted in 50 μL of $H_2O$, and samples were analyzed by WGS.

## Fitness assays

<u>In vitro:</u> $1 \times 10^5$ U-2 OS or C6/36 cells/well were seeded in 24-well plates, cultured for 24 h, and then triplicate wells were infected at an MOI of 0.1 FFU/cell with 1:1 mixtures of CHIKV WT or the indicated nsP4 mutant. The virus titer in the supernatant was measured by FFA 24 hpi, and the supernatant was then serially passaged on new cells two more times at the same MOI. RNA was extracted from the input virus mixture and passaged supernatants; cDNA was generated; and the frequency of mutations was analyzed by WGS. Results are shown for one experiment, with triplicate wells of each virus mixture. <u>In vivo:</u> C57BL/6 mice were inoculated in the

left footpad with a total of 1,000 FFU of a 1:1 mixture of CHIKV AF15561 WT (marked by introduction of an ApaI restriction site in the nsP4 region by silent mutation) and the indicated CHIKV nsP4 mutant in the AF15561 backbone. At 24 h post-inoculation, ipsilateral and contralateral ankle tissues were collected; RNA was extracted, cDNA was generated, and the nsP4 region of the viral genome was PCR amplified. PCR products were digested with ApaI (NEB) and PspOMI (NEB) at room temperature for 30 min followed by 2–3 h at 37˚C. The redundancy of the double digestion with the neoschizomers ApaI and PspOMI ensured complete digestion of the genetically marked PCR products. Digested PCR products were analyzed on a 1% TAE agarose gel, stained with ethidium bromide, imaged, and band intensities per lane were quantified (Syngene G Box). The percent band intensity is displayed.

## Mouse experiments

4-week-old male and female C57BL/6 mice were obtained from Jackson Laboratories. Eight mice per group (4 male; 4 female) were used for all studies. Mice were anesthetized with iso-flurane vapors and inoculated in the left rear footpad with a 10 µl volume of PBS/1% FBS containing $10^3$ PFU of CHIKV AF15561 or mutants made in the AF15561 backbone. Foot swelling, assessed using digital calipers, and weight measurements were taken daily after infection [68]. Tissue samples were collected at 7 days dpi. Tissues were homogenized in Trizol reagent (Life Technologies) and stored at -80˚C.

## Real-Time quantitative polymerase chain reaction (RT-qPCR) of mouse tissues

To quantify viral RNA in tissues, RNA was extracted from tissues homogenized in Trizol using a PureLink RNA Mini Kit (Invitrogen). During RNA isolation, samples were subjected to an on-column DNase treatment to eliminate DNA contamination. cDNA was generated from 1 µg of tissue-derived total RNA using random hexamer primers and SuperScript IV reverse transcriptase (Life Technologies). CHIKV copies were quantified using CHIKV-specific forward primer (5'-TTTGCGTGCCACTCTGG-3') and reverse primer (5'-CGGGTCACCACAA AGTACAA-3') with an internal TaqMan probe (5'-ACTTGCTTTGATCGCCTTGGTGAGA-3'), all within the nsP2 region of the genome, as previously described [69]. The total number of CHIKV genomes was extrapolated from a standard curve generated from samples containing $10^8$ to $10^0$ copies of CHIKV genomic RNA spiked into total RNA from BHK-21 cells, and cDNA was synthesized under conditions identical to those for samples from tissues. Samples were run and analyzed on a QuantStudio 7 Real-Time PCR system (Applied Biosystems).

## Statistical analysis

Statistical significance was assessed using one-way ANOVA or two-way ANOVA with multiple comparisons test in GraphPad Prism, version 10. The number of replicates per experiment and the p values are indicated in the figures and legends.

## Supporting information

**S1 Fig. Alignment of alphavirus nsP4 sequences around residues (A) Q192 or (B) C483.** Q192 and C483 are indicated by arrows, and the motifs in the nsP4 palm domain are marked by black boxes at the top of the sequences. The blue boxes show the conserved regions. TT stands for tight turn. Swiss-Prot accession numbers: CHIKV: A4L7I2, ONNV: P13886, MAYV: Q8QZ73, SFV: P08411, RRV: P13887, BFV: P87515, GETV: Q5Y389, SINV: P03317,

WEEV: P13896, VEEV: P36328, EEEV: Q306W6.
(TIF)

**S2 Fig. Alignment of nsP4 sequences across CHIKV strains around residues (A) Q192 or (B) C483.** Q192 and C483 are indicated by arrows. The blue boxes indicate conserved regions. TT stands for tight turn. GenBank numbers are indicated; the GenBank number for strain AF15561 is EF452493.
(TIF)

**S3 Fig. Alignment of alphavirus nsP2 sequences around residues H687 and K704.** H687 and K704 are indicated by arrows. The blue boxes indicate conserved regions. TT stands for tight turn. The Swiss-Prot accession numbers are the same as in S1 Fig.
(TIF)

**S4 Fig. Alignment of nsP2 sequences across CHIKV strains around residues H687 and K704.** H687 and K704 are indicated by arrows. The blue boxes indicate conserved regions. TT stands for tight turn. GenBank numbers are indicated; the GenBank number for strain AF15561 is EF452493.
(TIF)

**S5 Fig. Location of mutations in the nsP2 structure.** (A) Linear diagram of CHIKV nsP2 sequence, indicating the domains and the positions of the H687 and K704 residues. (B) Structure of the nsP2 protease and MTL domains, indicating the positions of H687 and K704. PDB:4ZTB [23].
(TIF)

**S6 Fig. The nsP2-K704N mutation does not alter fitness of the nsP4-C483Y mutant in U-2 OS or C6/36 cells.** (A and B) Competition assays comparing relative viral fitness in U-2 OS cells. Cells were inoculated and viruses serially passaged and analyzed as in Fig 4A. (C and D) Competition assays comparing relative viral fitness in C6/36 cells. Cells were inoculated and viruses serially passaged and analyzed as in Fig 4D. Panels A and C report the frequency of nsP4-C483Y to represent the nsP2-K704N+nsP4-C483Y mutant. Panels B and D report the frequency of nsP2-K704N to distinguish the two competitors.
(TIF)

**S1 Table. Summary of mutations with frequency >20% in CHIKV lineages.**
(DOCX)

**S2 Table. Summary of mutations in plaque-purified 4′-FIU-selected lineages.**
(DOCX)

**S3 Table. Primers used to generate mutations in CHIKV infectious cDNA clones.**
(DOCX)

**S1 Data. Source data for Fig 1.**
(XLSX)

**S2 Data. Source data for Fig 2.**
(XLSX)

**S3 Data. Source data for Fig 3.**
(XLSX)

**S4 Data. Source data for Fig 4.**
(XLSX)

**S5 Data. Source data for Fig 5.**
(XLSX)

**S6 Data. Source data for Fig 6.**
(XLSX)

**S7 Data. Source data for S6 Fig.**
(XLSX)

## Acknowledgments

We thank the Einstein Analytical Imaging Facility for use of their instruments, and the following facility staff for expert training and technical assistance: Vera DesMarais and Andrea Briceno. Facilities at Einstein were supported in part by the Cancer Center Core Support Grant NIH/NCI P30-CA013330. We thank Dr. Jonathan Lai at Einstein for the use of his Immuno-Spot S6 Macroanalyzer. EIDD-2749 was supplied by the Emory Institute for Drug Development (EIDD), Atlanta, Georgia, who also reviewed the manuscript.

## Author Contributions

**Conceptualization:** Peiqi Yin, Thomas E. Morrison, Margaret Kielian.

**Data curation:** Peiqi Yin, Elizabeth B. Sobolik.

**Formal analysis:** Peiqi Yin, Elizabeth B. Sobolik.

**Funding acquisition:** Thomas E. Morrison, Alexander L. Greninger, Margaret Kielian.

**Investigation:** Peiqi Yin, Elizabeth B. Sobolik, Nicholas A. May, Sainan Wang, Atef Fayed, Dariia Vyshenska, Adam M. Drobish, M. Guston Parks, Laura Sandra Lello.

**Methodology:** Peiqi Yin, Elizabeth B. Sobolik, Andres Merits, Alexander L. Greninger.

**Project administration:** Peiqi Yin, Andres Merits, Thomas E. Morrison, Alexander L. Greninger, Margaret Kielian.

**Resources:** Alexander L. Greninger.

**Supervision:** Thomas E. Morrison, Margaret Kielian.

**Visualization:** Peiqi Yin, Elizabeth B. Sobolik.

**Writing – original draft:** Peiqi Yin.

**Writing – review & editing:** Peiqi Yin, Elizabeth B. Sobolik, Andres Merits, Thomas E. Morrison, Alexander L. Greninger, Margaret Kielian.

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
