## [Decision Letter · Decision Letter 0]

13 Oct 2024

Dear Dr. Kielian,

Thank you very much for submitting your manuscript "Mutations in chikungunya virus nsp4 decrease viral fitness and sensitivity to the broad-spectrum antiviral 4′-Fluorouridine" for consideration at PLOS Pathogens. As with all papers reviewed by the journal, your manuscript was reviewed by members of the editorial board and by several independent reviewers. The reviewers appreciated the attention to an important topic. Based on the reviews, we are likely to accept this manuscript for publication, providing that you modify the manuscript according to the review recommendations.

The questions raised by reviewer one should be addressed with particularly careful consideration.

Sincerely,

George A. Belov, PhD

Academic Editor

PLOS Pathogens

Michael Letko

Section Editor

PLOS Pathogens

Michael Malim

Editor-in-Chief

PLOS Pathogens

orcid.org/0000-0002-7699-2064

Reviewer Comments (if any, and for reference):

Reviewer's Responses to Questions

**Part I - Summary**

Reviewer #1: In this manuscript, Yin et al. assess the potential of the ribonucleoside analog 4’Fluorouridine (4’-FIU) as effective antiviral against CHIKV by characterizing the in vitro resistance profile of CHIKV to the compound and testing whether the resistance mutations alter viral fitness in vitro and in vivo. CHIKV mutants less sensitive to 4’-FIU were selected during multiple rounds of passaging under 4’-FIU pressure in U-2 OS cells. Different sets of mutations were detected by Illumina whole genome sequencing, from which two pairs of mutations in nsP2 and nsP4 were most prevalent. Two mutations in nsP4 were responsible for the lower sensitivity to 4’-FIU, i.e., nsP4-C483Y and nsP4-Q192L. They then determined whether these mutations altered viral fitness in vitro and in vivo. NsP4-C483Y virus remained as fit as WT CHIKV, whereas the nsP4-Q192L mutation decreased viral fitness. Additionally, the authors performed RdRp fidelity analyses, because nsP4-C483Y was previously shown to alter polymerase fidelity. However, no significant differences between WT, nsP4-C483Y, and nsP4-Q192L were noted. Taken together, these data confirm previously published data on nsP4-C483Y regarding viral fitness and provide new insights on the novel mutation nsP4-Q192L. However, the most important conclusion in the manuscript is not supported by statistical analysis. Furthermore, the mutations nsP4-C483Y and nsP4-Q192L were characterized as stand-alone entities, whereas nsP4-Q192L was detected in conjunction with nsP2-H687P, which could potentially be a compensatory mutation. Hence, the manuscript should be reinforced with novel experimental data as outlined below.

Reviewer #2: The manuscript by Yin et al describes the selection of 4’-FlU resistant CHIKV variants and their genetic and phenotypic characterization, both in cell culture and a mouse model. Mutations in the RdRp nsP4 were responsible for resistance against the chain terminator 4’-FlU. Mutations in nsP2 had a minor contribution. The paper convincingly demonstrates that the C483Y mutation (or other nsp4 mutation) does not affect fidelity, i.e mutation frequency in contrast to what the first study on this mutation claimed. However, later studies already suggested there was no strong effect of C483Y on fidelity. Therefore, the findings are not novel and the current study confirms the lack of a significant effect of C483Y and provides solid additional evidence. The antiviral effect of 4’-FlU was described before in an elaborate study by the same group. The current study does not provide much new information on the mode of action of 4-FlU or the mechanism involved in 4’-FlU resistance. The study is well performed and the results support the claims and conclusions of the manuscript. The properties, fitness and in particular the mutation frequency of the mutants is extensively characterized. The manuscript is concise and clearly written.

Reviewer #3: In this manuscript, Yin, et al establish that mutations in CHIKV that confer diminished sensitivity to 4′-fluorouridine (4′-FlU) at low to moderate concentrations do not confer a fitness advantage in vitro or in vivo, thus adding to the evidence that (4′-FlU) could potentially be used as a clinically effective antiviral treatment against CHIKV. To establish this, the authors first serially passage the attenuated CHIKV strain 181/25 and assess the mutations that arise over the course of passaging. They find and characterize multiple mutations that arise in the nsP2 and nsP4 replicase proteins in terms of fitness, dsRNA production, and 4′-FlU sensitivity in both mammalian and mosquito cell culture. In vivo data using the AF15561 strain clearly demonstrates that while two dominant mutations lead to decreased sensitivity to 4′-FlU, these mutations do not confer a fitness advantage over wild type virus.

This reviewer found the overall work to be of good quality, generally well written, and would be of interest to readers of Plos Pathogens. However, there are several points, particularly related to reporting of methods used, that if addressed would improve the manuscript.

Outstanding questions:

- The AF15561 strain was isolated some time ago. Are there AA differences in nsP2 or nsP4 across CHIKV strains- especially currently circulating or more recent clinical isolates? In figures 1D and 1F nsP2 mutations seem to occur concurrently or after nsP4 mutations. Could preexisting variances in more recent/clinically relevant strains already have some of these nsP2 mutations that might alter the interpretation of the data in this manuscript?

Figure 1:

- Panel C was confusing due to the arrangement of the text above each Venn diagram. This should be adjusted so that the titles above each circle are easier to read and more closely associate with the circle they correspond to. The figure legend should also denote which colors represent the lineage mutations vs the plaque purified virus ones.

- The number of plaques that were purified from each lineage is missing. Please clearly state this.

Table S1:

- It would be helpful if there were a legend for this table

- What does the “depth” column indicate?

- For lineage 2: the AA change is listed as C489Y whereas in line 137 it is denoted as C483Y. Is this correct or is the nucleotide change of nsp4: G1466A correct?

- In line 138, please clarify “with the total frequency of these two nsP4 mutations at ~90%”. Do you mean that the sum of the frequency of having C483Y or Q192L was ~90%?

Figure 2:

- Panel C and D- Line 172 states that a “low MOI” was used but the figure legend states an MOI=10. Please clarify.

- Please indicate what the early (pre-12hr) time points collected were. This is not stated or listed on graph.

Figures 3 and 4 were swapped. Figures will be referenced here as they were indicated in the text, not the uploaded figures.

Figure 3:

- The schematic in panel B does not match what is stated in the methods (lines 430-432). Was a 2 or 3 plasmid system used for these experiments? Additionally, the timing indicated in the legend doesn’t match what is stated in the methods.

Figure 4:

- Ideally, it would have been nice to see at least one more biological replicate, but at minimum, please state what is plotted in these graphs. Are the % frequencies the averages of the technical replicates?

- The restriction digestion technique used for panels G and H needs substantial clarification. It was hard to make sense of how the data was generated and analyzed from the description in the text. A diagram and the inclusion of raw data, at least as an example in a supplemental figure, would be helpful in better evaluating these experiments.

Figure 5:

- For panel A, please define what overall mutation frequency is and if there was, for example, a threshold set (ex 80% or higher) to determine what was counted as a mutation. It is difficult to understand how there could be more mutated sites than mutations.

- For panel C and line 237, please define Shannon entropy, how it was determined or calculated, and what it tells us. This was not specified anywhere in the text.

- The presentation of data in panel D was confusing and needs to either be shown in a different way or more thoroughly explained in the text. The figure key was also confusing. “>1%” of what? Should this also be >1 and <5, since all the >5 are also >1. Also, please move data reporting in legend (lines 868-869) to the results section.

- Please specify how statistics were done for panels E and F. Please define what * means in terms of significance in the legend.

Figures S1 and S2:

- These figures need legends describing what the features are. What are the blue and black boxes? Please label or define in the legend what the numbers at the top are. What does TT stand for? Could the arrows and labeling of motifs be adjusted to be clearer which is which? It is hard to decipher. Also please indicate the strains of viruses from which the sequences were obtained.

Additional textual edits:

- Line 196- “synthesis of the full length” was confusing. Both RNAs are going to be “full length”.

- Line 285- “the CHIKV replication complex does not contain viral RNA”- please make it more clear that this referencing the depicted structure and not the replication complex itself, which clearly contains viral RNA.

- Line 295- while the motif D may not be conserved, it seems that C483 is completely conserved (Fig. S1). Please alter text accordingly.

**Part II – Major Issues: Key Experiments Required for Acceptance**

Reviewer #1: The conclusion that nsP4 mutations nsP4-C483Y and nsP4-Q192L are primarily responsible for the reduction in sensitivity to 4’-FIU is based on the dose-response curves in Fig.2A+B, but no statistical analysis was performed whilst the variation is visibly high. The authors should perform statistics on the EC90 analysis and show the individual data points in Fig.2B. If the differences are not significant due to the high variation, the authors should amend the main text to phrase the results in a less strong fashion. Has the same set of nsP2 and nsP4 mutants been included in the trans-replication assay (Fig. 3C,D) and the dose-response curve in human dermal fibroblasts (Fig. 6A), although not shown in the manuscript? Please, include the whole set of mutants (from Fig.2) in these two assays to support the main conclusions from the manuscript by orthogonal methodology.

Viral fitness of nsP4-C483Y and nsP4-Q192L was assessed in growth curves and competition assays in vitro and in vivo. What was the reason why the nsP4 mutants were characterised as stand-alone entities and not in context of the other mutations detected by whole genome sequencing? NsP4-Q192L, which results in a reduction in viral fitness, has only been detected in conjunction with nsP2-H687P. The effect of nsP2-H687P on the viral fitness of nsP4-Q192L should be assessed (at least in vitro) to confirm or negate that nsP2-H687P is a compensatory mutation restoring viral fitness of nsP4-Q192L to WT levels.

Although the manuscript is written comprehensively, it is not supported by intuitive and to-the-point figures. The figures contain too much white space, whereas titles of graphs and legends often contain redundant information (for example, adding CHIKV to all figure legends is unnecessary when it is the only virus studied). The authors should streamline the figures and resolve any inconsistencies, see detailed remarks in the Minor Comments section.

It is unclear how the trans-replication assay was performed. Figure 3B clearly shows that two plasmids (nsPs + template) were transfected, whereas the Material and Methods states that nsP4 was transfected on a third separate plasmid.

Consider placing the section on RdRp fidelity after the section on nsP4 activity (switching Figures 4 and 5). This would divide the manuscript in ‘effect on RdRp function’ and ‘viral fitness assessment’, enhancing its clarity for the reader.

Is there any evidence that 4’-FIU selects for the here-described CHIKV mutations in vivo during 4’-FIU treatment? Has WGS been performed on the 4’-FIU-treated mice inoculated with WT CHIKV or is this data available from the previously published study initially describing the effectivity of 4’-FIU as CHIKV antiviral (doi: 10.1128/mbio.00420-24)?

Reviewer #2: (No Response)

Reviewer #3: (No Response)

**Part III – Minor Issues: Editorial and Data Presentation Modifications**

Reviewer #1: • Title: capitalize the P in nsP4.

• Line 67: assemble ‘as’ progeny virions.

• Line 92: add that ribavirin is a nucleoside analogue.

• Line 94: please rephrase ‘which has been shown to have effects on RdRp fidelity’, see later comment about lines 227-228.

• Line 103: what is meant by ‘the’ physiological nucleotide, is there a specific one it mimics?

• Line 116: add a nuanced reasoning to this conclusion, because the authors do show that CHIKV develops resistance to 4’-FIU, so why is it still a good candidate?

• Line 121: add that U-2 OS is a human osteosarcoma cell line.

• Line 129: add that the MOI of the initial primary stock was 0.5 as well.

• Line 130: add that just one virus clone was plaque purified and sequenced per lineage.

• Line 132: add that WGS consisted of Illumina deep sequencing to distinguish it from nanopore sequencing.

• Line 140: which threshold was used to define ‘high’?

• Lines 133-159: Consider rewriting this section in a more concise manner.

• Fig 1:

o Panel A: add that one plaque was purified and sequenced.

o Panel B: match the connecting line of the DMSO lineages to the symbol (do not use white). Add space between lineage and number. ‘MOI’ is not aligned.

o Panel C: typo, it says nsP4-Q912L instead of 192.

o Panels E, G, I: simplify by choosing one colour and show a gradient for the passages.

o Figure legend: add (red) and (blue) to section describing panel (C). Add space between lineage and number (lines 801 and 803).

• Line 163: ‘resulted in’ instead of ‘produced’.

• Line 166: ‘and’ instead of ‘or’.

• Line 168: as mentioned as Major Comment, rephrase the conclusions with more nuance because no statistics has been performed.

• Fig 2:

o Panel B: add individual data points and statistics

o Panel C, D: The Y axis title does not state FFU/mL, has another method been performed that has not been described in the Material and Methods? Change the X axis title to Hours Post Inoculation (be consistent in all figures and main text).

o Panel E: The N and C termini are not aligned.

o Figure legend: add (nsP2h) in between brackets after ‘nsP2 helicase domain’.

• Line 194: rephrase, a mutant does not have the ability to decrease something, the mutant results in a decreased sensitivity

• Line 194: change ‘precursor’ to ‘polyprotein’ as the polyprotein can already be active prior to nsP2 cleavage.

• Lines 196-199: extend the explanation on what Fluc and Gluc activities represent within the context of the trans-replication assay. Gluc activity is not a sole read-out of ‘transcription’ because genome replication is required prior to SG RNA production.

• Fig 3:

o Panel A: add DAPI as separate panel, because it is present in the merged image. dsRNA is capitalized whereas nsP4 is not.

o Panel B: see Major Comment, does not correspond to Materials&Methods.

• Line 218: add one sentence on how the ratio was determined using the ApaI restriction site.

• Lines 227-228: change the introduction (line 94) to include the contrasting results on the effect of C483Y on RdRp fidelity mentioned here.

• Line 236: the word frequency makes this sentence confusing, use ‘number’ instead.

• Fig 5: reduce the amount of white space.

o Panel B: the Y axis title seem wrong as it is the same as in panel A.

o Panel D: change legend to 1-5% and 5-10% instead of using the symbol ‘>’. Match colors to panels A-C.

• Line 259: add ‘was determined by RT-qPCR’.

• Fig 6: To reduce white space, remove all CHIKV-labels in figure legend titles.

o Panels C, D: change X-axis title from ‘Days post infection’ to ‘Days post inoculation’ to match main text and other figures.

o Panel E: change u to μ. Change the 10 in log10 to subscript.

• Line 285: add that this sentence refers to the reconstituted replication complex.

• Line 313: what is meant by ‘isolation’? Please rephrase.

• Line 326-327: it is already clear that the finger domain is important for replication. Please, rephrase.

• Line 370: add antibody concentrations.

• Line 382: change ‘prepared’ to ‘RNA was isolated’. Add whether the stocks were frozen in between the passages.

• Line 388: typo ‘cycles’.

• Line 406: change ‘in alignment with’ to ‘similar to’.

• Lines 422-428: which medium was used for the IF infections and were the cells washed after 1 hour?

• Line 425: add antibody concentrations.

• Lines 441-445: Were UMIs added to the library to increase the accuracy of the fidelity analysis and if not, why?

Reviewer #2: 1 The section of the results that starts at ln 177 is more discussion than results.

2 The study is done with the attenuated vaccine strain 181/25 or the asian lineage strain AF15561, which is less pathogenic and exhibits decreased replication compared to other lineages like ECSA. Could the authors discuss this limitation. Could the use of these strains explain the lack of an effect on fidelity (for C483Y) in this study? As the first study that reported increased fidelity was done with an ECSA isolate. Authors could check the effect of C483Y in the ECSA background with their assays and compare it to the 181/25 results.

3 Do the authors have an explanation why the resistance is not increasing over ~5-fold? Have they tried longer passaging and higher doses of compound? Did extra mutations emerge?

4 Fig 1D - Can it be concluded that the early Q192P mutant was outcompeted by the C483Y mutant? Is there experimental evidence for this, like for Q192L?

5 Minor point: Fig 1C – nsP4-Q192L mutation listed in figure as Q912L

Reviewer #3: Please see I above.

PLOS authors have the option to publish the peer review history of their article (what does this mean?). If published, this will include your full peer review and any attached files.

Reviewer #1: No

Reviewer #2: No

Reviewer #3: No

Figure Files:

Data Requirements:

Reproducibility:

References:

---

## [Editor Report · Decision Letter 1]

23 Dec 2024

Dear Dr. Kielian,

We are pleased to inform you that your manuscript 'Mutations in chikungunya virus nsp4 decrease viral fitness and sensitivity to the broad-spectrum antiviral 4′-Fluorouridine' has been provisionally accepted for publication in PLOS Pathogens.

Best regards,

George A. Belov, PhD

Academic Editor

PLOS Pathogens

Michael Letko

Section Editor

PLOS Pathogens

Sumita Bhaduri-McIntosh

Editor-in-Chief

PLOS Pathogens

orcid.org/0000-0003-2946-9497

Michael Malim

Editor-in-Chief

PLOS Pathogens

orcid.org/0000-0002-7699-2064
---

## [Editor Report · Acceptance letter]

3 Jan 2025

Dear Dr. Kielian,

We are delighted to inform you that your manuscript, "Mutations in chikungunya virus nsp4 decrease viral fitness and sensitivity to the broad-spectrum antiviral 4′-Fluorouridine," has been formally accepted for publication in PLOS Pathogens.

Best regards,

Sumita Bhaduri-McIntosh

Editor-in-Chief

PLOS Pathogens

orcid.org/0000-0003-2946-9497

Michael Malim

Editor-in-Chief

PLOS Pathogens

orcid.org/0000-0002-7699-2064